# Phosphatidylethanolamine *N*-Methyltransferase Knockout Modulates Metabolic Changes in Aging Mice

**DOI:** 10.3390/biom12091270

**Published:** 2022-09-09

**Authors:** Qishun Zhou, Fangrong Zhang, Jakob Kerbl-Knapp, Melanie Korbelius, Katharina Barbara Kuentzel, Nemanja Vujić, Alena Akhmetshina, Gerd Hörl, Margret Paar, Ernst Steyrer, Dagmar Kratky, Tobias Madl

**Affiliations:** 1Research Unit Integrative Structural Biology, Division of Molecular Biology and Biochemistry, Gottfried Schatz Research Center for Cell Signaling, Metabolism and Aging, Medical University of Graz, 8010 Graz, Austria; 2Key Laboratory of Gastrointestinal Cancer, Fujian Medical University, Ministry of Education, Fuzhou 350122, China; 3Gottfried Schatz Research Center for Cell Signaling, Metabolism and Aging, Molecular Biology and Biochemistry, Medical University of Graz, 8010 Graz, Austria; 4Otto-Loewi Research Center, Division of Medicinal Chemistry, Medical University of Graz, 8010 Graz, Austria; 5BioTechMed-Graz, 8010 Graz, Austria

**Keywords:** metabolomics, NMR, PEMT, knockout, aging, mice, liver, intestine, white/brown adipose tissue

## Abstract

Phospholipid metabolism, including phosphatidylcholine (PC) biosynthesis, is crucial for various biological functions and is associated with longevity. Phosphatidylethanolamine *N*-methyltransferase (PEMT) is a protein that catalyzes the biosynthesis of PC, the levels of which change in various organs such as the brain and kidneys during aging. However, the role of PEMT for systemic PC supply is not fully understood. To address how PEMT affects aging-associated energy metabolism in tissues responsible for nutrient absorption, lipid storage, and energy consumption, we employed NMR-based metabolomics to study the liver, plasma, intestine (duodenum, jejunum, and ileum), brown/white adipose tissues (BAT and WAT), and skeletal muscle of young (9–10 weeks) and old (91–132 weeks) wild-type (WT) and PEMT knockout (KO) mice. We found that the effect of PEMT-knockout was tissue-specific and age-dependent. A deficiency of PEMT affected the metabolome of all tissues examined, among which the metabolome of BAT from both young and aged KO mice was dramatically changed in comparison to the WT mice, whereas the metabolome of the jejunum was only slightly affected. As for aging, the absence of PEMT increased the divergence of the metabolome during the aging of the liver, WAT, duodenum, and ileum and decreased the impact on skeletal muscle. Overall, our results suggest that PEMT plays a previously underexplored, critical role in both aging and energy metabolism.

## 1. Introduction

Aging affects the physiology of living organisms, causes a multitude of defects in biological functions, and is induced by orchestrated alterations in various metabolic pathways [1,2,3]. Systematic studies, particularly during the past decades, have investigated the biological features of aging at the cellular, transcriptomic, proteomic, metabolomic, and molecular levels [4,5,6,7,8,9] and have linked a remarkable number of biological mechanisms to aging. Dysregulated lipid metabolism contributes to aging via various complex mechanisms [10,11,12]. Newly emerging data are indicating that the biosynthesis of phospholipids (PLs), triglycerides, fatty acids, and sphingolipids plays a key role in lifespan regulation [13,14,15]. Notably, PLs are the major components of membranes in eukaryotic cells [16,17], and their association with aging has been demonstrated recently [18]. Phosphatidylcholine (PC) and phosphatidylethanolamine (PE) are the two most abundant PLs in mammalian cells and are important building blocks for more complex PLs [14,19,20]. They also play critical roles in processes such as the biosynthesis of biological membranes, lipid transport via lipoproteins, lipid droplet formation [21], the regulation of mitochondrial function, cell proliferation/apoptosis, insulin signaling, and whole-body energy homeostasis [19,22,23].

The biosynthesis of PC from PE in mammals is catalyzed by phosphatidylethanolamine *N*-methyltransferase (PEMT) via three sequential steps of PE methylation by transferring the methyl group from S-adenosylmethionine to the corresponding substrates, i.e., PE, phosphatidylmonomethylethanolamine, and phosphatidyldimethylethanolamine [24,25,26]. This process mediates about 30% of liver PC biosynthesis and is one of three PC biosynthesis pathways in mammalian cells [27,28]. PEMT1 and PEMT2 are the two isoforms of PEMT, with PEMT1 localized in the endoplasmic reticulum (ER) and PEMT2 in the mitochondria-associated membranes (MAM) [29,30]. They are encoded by the same gene and perform the same enzymatic activities [31]. The other two biosynthetic pathways of PC are the cytidine-5′-diphosphocholine (CDP-choline) pathway, which adds CDP-choline to diacylglyceride, and the Lands cycle, which adds a fatty acid to lysophosphatidylcholine [27,32]. The PEMT pathway is therefore the only pathway that synthesizes PC independent of dietary choline or its derivatives, and the disruption of the PEMT pathway in combination with a choline-deficient diet is lethal to knockout mice [30]. PEMT has a prominent role in the secretion of very low-density lipoproteins (VLDL) by the liver [19,33,34], and regulates the formation of lipid droplets (LD) by adipocytes [21]. Furthermore, the PEMT and CDP-choline pathways have been found to yield PCs with fatty acids of different lengths and degrees of unsaturation in yeast cells and rat hepatocytes, which may underline the importance of the respective pathways [35,36]. Additionally, the extent to which the PC concentration changes in the liver of PEMT KO mice remains unclear. Controversial results have been found regarding the hepatic PC concentration under PEMT deficiency [31,37], and an additional study indicated a lower mitochondrial PC level in the KO liver [38]. Considering both the subcellular enrichment of PEMT on ER and MAM [21,30], both the total and subcellular PC levels in the KO liver remain elusive. PEMT deficiency leads to various biological changes, including increased hepatic steatosis [39], attenuated ER stress in diabetic nephropathy [40], a resistance to diet-induced obesity, and protection against insulin resistance in mice [41,42,43]. These observations suggest a unique role for PEMT in lipid metabolism.

It is unclear whether the methylation of PE by PEMT regulates the progression of aging because the levels of both substrate and product, i.e., PE and PC, are found to either increase or decrease in various aged tissues from humans, mice, or nematodes [18]. Interestingly, the dietary supplementation of both PE and PC has been found to contribute to lifespan extension in *C. elegans* through the same mechanism that reduces insulin/IGF-1-like signaling and promotes DAF-16 activity [44,45,46]. Several unsaturated PCs play a protective role against neurodegenerative diseases in humans [18]. A decrease in PE levels during aging has been observed in *C. elegans* and mice [47,48], whereas an increase in PE levels prolonged the lifespans of yeast and flies and increased the replicative viability of mammalian cells [49]. While PEMT is of particular importance for lipid metabolism and membrane biosynthesis, a systematic investigation of the molecular basis of its function is required to elucidate the additional functions of PEMT beyond its function in PC synthesis.

Mouse models are well suited to the study of age-related metabolic changes due to their analogy to human metabolism [50,51,52]. Herein, we investigated the role of PEMT in lipid metabolism and energy homeostasis in young (9–10 weeks) and old (96–104 weeks) wild-type (WT) and PEMT knockout (KO) mice. Using untargeted NMR-based metabolomics, we assessed the metabolic alterations caused by PEMT deficiency in the liver, intestine (duodenum, jejunum, and ileum), plasma, brown and white adipose tissues (BAT and WAT), and skeletal muscle. The liver is the tissue with by far the highest expression level of PEMT [53]. We expected to observe the direct consequence of PEMT KO on the metabolism of the aforementioned tissues, especially the liver, as well as the homeostasis of PC in the corresponding tissues to compensate for the lack of PEMT-derived PC. Furthermore, we aimed to identify the indirectly linked metabolic alterations in these tissues in response to PEMT deficiency, including potential secondary or remote effects.

We addressed the identification of changes in the metabolome through a pairwise comparison of WT and KO tissues, or young and aged tissues, respectively. Most KO tissues displayed moderate changes in their metabolomes. Notably, the metabolome of the BAT from both young and old mice, as well as the liver and ileum from young mice, were dramatically affected by the loss of PEMT. Associated with this, we identified a large number of significantly altered metabolites in each of these tissues. The impact of PEMT deficiency on the metabolome of the liver and BAT appeared to be largely age-independent, suggesting that these are the core tissues driving the metabolic changes in KO mice. As for their aging profiles, we identified metabolites as aging biomarkers that were either increased or decreased in the KO tissues. All KO tissues had altered sets of aging biomarkers compared with their WT counterparts. PEMT-deficient skeletal muscle displayed fewer aging biomarkers than WT skeletal muscle. The aging profiles of plasma and BAT remained partly PEMT-independent. Notably, the aging metabolomes of the WAT, duodenum, and ileum from KO tissues were associated with more aging biomarkers than the WT tissues. This suggested that their metabolic adaptations to the lack of PEMT are highly affected by aging, resulting in a strong add-on effect. The jejunum remained metabolically unchanged both during aging and PEMT KO, demonstrating its metabolic stability. These results indicated that PEMT plays a critical role in energy metabolism by regulating the function of most associated tissues and their functional changes during aging.

## 2. Materials and Methods

### 2.1. Animals

Mice globally lacking PEMT were generated in the laboratory of Dennis E. Vance on a mixed genetic background (129/J and C57BL/6J). Young (9–10 weeks of age) and old (96–104 weeks of age for WT, and 91–132 weeks of age for PEMT KO) female mice were housed in a temperature-controlled environment with *ad libitum* access to food (standard chow diet; 11.9% caloric intake from fat; Altromin, Lage, Germany) and water in a regular light–dark cycle (12 h/12 h). Organs were collected after overnight fasting. All experiments were performed in accordance with the European Directive 2010/63/EU. Metabolomic studies of several tissues of aged WT mice have been published previously [52].

### 2.2. Pretreatment of NMR Samples

From each tissue, 30–50 mg was isolated, snap-frozen in liquid nitrogen, and stored at −80 °C until metabolite extraction. To isolate metabolites, 400 μL of ice-cold methanol and 200 μL of MilliQ H_2_O were added to each tissue sample, while 400 µL of ice-cold methanol was added to 200 µL plasma sample. Tissue homogenization was performed using the Precellys 24 tissue homogenizer (Bertin Technologies, Montigny-le-Bretonneux, France) in 2 mL tubes with O-ring caps and Precellys beads (1.4 mm zirconium oxide beads, Bertin Technologies, Villeurbanne, France). The homogenized tissues were centrifuged at 13,000 rpm for 45 min (4 °C), after which the supernatants were transferred into new 1.5 mL tubes and lyophilized at <1 Torr, 850 rpm, and 25 °C for 10 h in a vacuum-drying chamber (Savant Speedvac SPD210 vacuum concentrator) with an attached cooling trap (Savant RVT450 refrigerated vapor trap) and vacuum pump (VLP120) (all Thermo Scientific, Waltham, MA, USA). The lyophilized samples were subsequently dissolved in 500 μL NMR buffer (0.08 M Na_2_HPO_4_, 5 mM TSP (3-(trimethylsilyl) propionate-2,2,3,3-d4 sodium salt, 0.04 (*w*/*v*)% NaN_3_ in D_2_O; pH adjusted to 7.4 with 8 M HCl and 5 M NaOH) for NMR analysis. For lipid removal, 50 µL of chloroform (CHCl_3_) was added to the tissue lysates of BAT, WAT, and intestine and centrifuged at 13,000 rpm for 10 min (4 °C). The supernatants were transferred to 5 mm NMR tubes for data collection.

### 2.3. Data Acquisition and Analysis

Metabolic profiling was conducted at 310 K using a 600 MHz Bruker Avance Neo NMR spectrometer (Bruker Biospin, Rheinstetten, Germany) equipped with a TXI 600S3 probe head. ^1^H 1D NMR spectra were recorded using Carr–Purcell–Meiboom–Gill (CPMG) pulse sequence with a pre-saturation for water suppression (cpmgpr1d, 512 scans, 73,728 points in F1, 12019.230 Hz spectral width, 1024 transients, recycle delay 4 s) [54,55], and ^1^H-^13^C heteronuclear single-quantum correlation (HSQC) spectra were recorded with a recycle delay of 1.0 s, spectral widths of 20.8/83.9 ppm, and centered at 3.9/50.0 ppm in ^1^H/^13^C with 2048/256 points, respectively, and 8 scans per increment. NMR spectral data were processed in Bruker Topspin version 4.0.2 using one-dimensional exponential window multiplication of the FID, Fourier transformation, and phase correction as previously described [56]. TSP was used as the internal standard for chemical-shift referencing (set to 0 ppm), and processed NMR data were imported into MATLAB R2014b. Signal suppressions were applied to regions corresponding to the ^1^H chemical shift of water, TSP, and methanol (and CHCl_3_ for BAT, WAT, and intestines, as well as EDTA for plasma). Due to field homogeneity issues, some samples (mainly BAT and WAT) had to be excluded from the data analysis. Alignment and a probabilistic quotient normalization were performed on the ^1^H 1D NMR spectra by MATLAB R2014b to generate normalized spectra and reduced spectra. Metabolite quantification was based on the signal integration of normalized spectra as described previously [52]. Signal integration was used to generate a volcano plot, orthogonal partial least squares discriminant analysis (O-PLS-DA), permutation analysis, sparse partial least squares discriminant analysis (S-PLS-DA), metabolite set enrichment analysis (MSEA; only over representation analysis (ORA) have been performed here [57]), and a heat map using MetaboAnalyst 5.0 [58]. Statistical significance was tested by quality assessment statistic Q^2^. A univariate statistical analysis was carried out for quantified metabolites using GraphPad Prism 7.04 (GraphPad Software, La Jolla, CA, USA). For plotting the statistical analysis in the Appendix A, data were shown as mean ± standard deviation (SD). An unpaired two-tailed Student’s *t*-test was used to calculate *p*-values for comparison of variables. Metabolites with *p* < 0.05 are shown in panel B and panel D of Appendix A. For plasma, skeletal muscle, adipose tissues and intestine segments, an additional 2D ^1^H-^13^C HSQC spectrum was recorded to highlight the presented metabolites and to validate the interpretations of 1D ^1^H spectra. These 2D spectra have been either published previously (for WT [52]) or included in Appendix A.

## 3. Results

A comparison of the individual metabolites revealed a set of increased and decreased metabolites for each tissue. These characteristic metabolic fingerprints partly diverged from the biomarkers of aging and manifested as tissue-specific profiles.

### 3.1. The Liver Metabolome Strongly Responds to the Absence of PEMT

The livers of young mice were more sensitive to PEMT deficiency than the livers of aged mice (Figure 1A,C). In young PEMT KO mice, this was associated with an increase in the concentrations of AMP, histidine, acetate, UDP-sugars, inosine, ADP, bile acids, glutamine, propionate, and glutathione, and a decrease in the concentration of mannose, hypoxanthine, choline, phosphorylcholine, phenylalanine, glycine, glycerol, and glucose. Among these metabolites, the concentrations of AMP (5.6-fold), histidine (3.1-fold), and UDP-glucose (2.1-fold) were increased by >2-fold in the KO mice compared to the WT mice, whereas the level of mannose was decreased by 2.1-fold. The concentration changes of other metabolites were <2-fold (Figure 1A, Appendix A). The O-PLS-DA yielded a score of 33.2% for the predictive component (T score [1]) and resulted in a good separation of the two groups, associated with an R^2^X of 0.332, an R^2^Y of 0.977, and a Q^2^ at 0.867, indicating the predictive accuracy of the O-PLS-DA model (Figure 1B). We further performed a permutation test and obtained correlation coefficients for R^2^Y up to 0.997 (*p* < 0.01) and a Q^2^ up to 0.915 (*p* < 0.01), which suggested a significant distinction between the two groups (Figure 1E).

In the hepatic metabolome of aged KO mice, we observed an increase in the levels of histidine (8.0-fold), AMP (6.0-fold), inosine (2.3-fold), acetate, propionate, and glutathione (<2-fold when not specified), as well as a decrease in the levels of fumarate (2.3-fold), phosphorylcholine, choline, and UDP-N-acetylglucosamine (Figure 1C, Appendix A). The O-PLS-DA yielded a T[1] score of 20.7% for the predictive component (Figure 1D). The permutation test additionally yielded correlation coefficients R^2^Y of 0.996 (*p* < 0.01) and a Q^2^ of 0.946 (*p* < 0.01), indicating that the groups can be discriminated with a high accuracy using the O-PLS-DA model.

A graphical representation of the four liver groups (S-PLS-DA) yielded a component 1 of 26.1% and a component 2 of 15.6%. All four groups clustered and separated well from each other (Figure 1G). The PEMT KO livers shared multiple aging-related biomarkers with the WT livers, including lactate, niacinamide, fumarate, alanine, inosine, and aspartate (Appendix A and previously published results [59]). To investigate the biological relevance of the metabolic changes, we eventually performed an over-representation analysis (ORA), an enrichment analysis of the metabolite set enrichment analysis (MSEA) [57], with the aim of highlighting the biological pathways associated with altered metabolite levels. In the young and aged livers, 23 and 11 metabolic pathways, respectively, were associated with the changes, with the 7 major pathways being galactose metabolism, ammonia recycling, lactose synthesis, glutamate metabolism, purine metabolism, PC biosynthesis, and alanine metabolism in young livers (Figure 1H), as well as aspartate metabolism, PC biosynthesis, pyruvate metabolism, ethanol degradation, phenylalanine and tyrosine metabolism, purine metabolism, and PL biosynthesis in aged livers (Figure 1I).

### 3.2. Aging Aggravates the Effects of PEMT Deficiency in Plasma

In the plasma of young mice, PEMT deficiency resulted in a 2.9-fold increase in the uremic metabolite indoxyl sulfate (Figure 2A, Appendix A) without affecting the abundance of other metabolites. The O-PLS-DA yielded a consistent score of 13.4% for the predictive component (T[1]), in association with a low degree of separation (Figure 2B). The permutation test resulted in an R^2^Y of 0.979 (*p* = 0.01) and a Q^2^ of 0.648 (*p* = 0.02), which indicated the significant difference of the two groups (Figure 2E).

In the plasma of the aged KO mice, the volcano plot showed a less than 2-fold increase in the levels of isoleucine, lactate, glucose, alanine, and glycerol, accompanied by decreased levels of citrate, myo-inositol, indoxyl sulfate, isobutyrate, and creatine, among which citrate decreased more than 3-fold and myo-inositol decreased to 2.2-fold (Figure 2C, Appendix A). The T[1] of the predictive component of the O-PLS-DA plot was 24.3%, which is consistent with a good predictive accuracy due to the distinct clustering of the two groups (Figure 2D). The permutation test yielded an R^2^Y of 0.902 (*p* = 0.02) and a Q^2^ of 0.731 (*p* = 0.01) (Figure 2F).

Plotting the four plasma groups on the S-PLS-DA plot resulted in a component 1 at 30.8% and a component 2 at 18.2% and yielded a distinct separation of WT young and old and between old WT and KO mice (Figure 2G). The aging profile of plasma from WT and PEMT KO mice have respectively 11 and 13 increased and decreased metabolites, 4 of which overlap, suggesting that PEMT affects the metabolic aging of plasma (Appendix A [52]). The ORA showed that metabolites with altered levels in the plasma of aged KO mice were related to five metabolic pathways, including galactose metabolism, the glucose–alanine cycle, the Warburg effect, the transfer of acetyl groups into mitochondria, and gluconeogenesis (Figure 2H).

### 3.3. Aging and PEMT Deficiency Affect Skeletal Muscle Metabolites in Opposite Ways

It has been previously shown that PEMT KO mice exhibit an increased energy expenditure along with an elevated basal oxygen consumption rate in skeletal muscles [60]. In line with this, we observed metabolic changes in the skeletal muscles of both young and aged PEMT KO mice compared to the corresponding WT animals (Figure 3A,C). As demonstrated in the volcano plot, young muscle tissues showed around 1.5-fold higher concentrations of acetate and reduced concentrations of inosine, malonate, phenylalanine, valine, and glutamate (Figure 3A, Appendix A), consistent with the score of the first predictive component at 27.3% on the O-PLS-DA plot (Figure 3B). The separation of the two groups was significant, as demonstrated by a permutation test yielding an R^2^Y of 0.986 (*p* = 0.01) and a Q^2^ of 0.825 (*p* = 0.01) (Figure 3E).

The quantification of the metabolites in the skeletal muscle samples from old mice showed an increase in acetate, glycine, and succinate levels, along with a decrease in inosine, malonate, leucine, lysine, and phenylalanine concentrations (Figure 3C, Appendix A). Regarding the O-PLS-DA, a T[1] score at 19.9% was obtained (Figure 3D). The significant separation of the two groups was further emphasized by the permutation test with a correlation coefficient, R^2^Y, of 0.985 (*p* < 0.01) and a Q^2^ of 0.715 (*p* < 0.01) (Figure 3F).

The S-PLS-DA plot showed a distinct separation of all four groups (Figure 3G), with different components separating samples from young/old and WT/KO mice. In addition, only 3 of the 15 biomarkers of aging skeletal muscle were conserved in the skeletal muscle of PEMT KO mice (Appendix A and reference [52]). This suggests that the aging of skeletal muscle is regulated by PEMT but is not fully PEMT-dependent. The ORA showed that metabolites with altered abundance in young KO skeletal muscle were associated with six pathways: aspartate metabolism, phenylalanine and tyrosine metabolism, amino sugar metabolism, fatty acid biosynthesis, propanoate metabolism, and valine, leucine, and isoleucine degradation (Figure 3H). Metabolites from aged samples were associated with three pathways: carnitine synthesis, aspartate metabolism, and fatty acid biosynthesis (Figure 3I).

### 3.4. PEMT Deficiency Strongly Affects the Metabolic Profile of Brown Adipose Tissue

Among the eight specimens we examined in this study, BAT is one of the most metabolically altered tissues in the absence of PEMT. In the volcano plot representing the metabolites detected in the BAT of young mice, we observed an increase in inosinate (>5-fold), citrate, inosine, methionine, and creatine, as well as a decrease in tryptophan and UDP-sugars (>8-fold), hypoxanthine, valine, alanine, lactate, histidine, isoleucine, acetate, glutamate, glycerophosphocholine, uridine, leucine, phenylalanine, niacinamide, mannose, and uracil (Figure 4A, Appendix A). A discriminant clustering of the two groups on the O-PLS-DA resulted in a score of the first predictive component of 46.7%, and the significance of their separation was validated by a permutation study with an R^2^Y of 0.997 (*p* = 0.02) and a Q^2^ of 0.970 (*p* = 0.02) (Figure 4B,E).

We found a similar distinction when comparing old WT and PEMT KO mice, yielding a concentration increase in inosinate (≈15-fold), fumarate, and threonine, accompanied by a decrease in tryptophan (≈8-fold), UDP-sugars, histidine (≈4-fold), acetate, formate, glycerophosphocholine, isoleucine, valine, phenylalanine, leucine, 3-hydroxybutyrate, glutamine, tyrosine, ethanolamine, and glutamate (Figure 4C, Appendix A). We found that 11 of the 22 metabolites that are either increased or decreased in the young KO’s BAT were found to change in a comparable trend in the aged KO BAT (Appendix A). The O-PLS-DA showed a clustering with a T[1] score of 34.6%, whereas the permutation validation yielded an R^2^Y of 0.986 (*p* < 0.01) and a Q^2^ of 0.892 (*p* < 0.01) (Figure 4D,F).

Notably, the S-PLS-DA plot showed a clear separation of the four groups, especially between the WT and KO groups. (Figure 4G). In the aging profile, 8 of the 14 aging biomarkers found in the WT tissues were also conserved in the KO tissues. These biomarkers are highlighted in the Venn diagram and are labeled on the volcano plot corresponding to the aging of KO BAT (Appendix A, [52]). This suggests an essential role of PEMT in BAT metabolism, which is partially aging-independent. We further performed an ORA using altered metabolites from the young and old BAT. The metabolites identified in the young tissues were related to the glucose–alanine cycle, glycine and serine metabolism, beta-alanine metabolism, valine, leucine and isoleucine degradation, aspartate metabolism, and alanine metabolism (Figure 4H). The metabolites from the aged tissues were related to aspartate metabolism, phenylalanine and tyrosine metabolism, the urea cycle, ammonia recycling, valine, leucine and isoleucine degradation, amino sugar metabolism, and purine metabolism (Figure 4I). This finding further demonstrates that PEMT manifests both aging-dependent and -independent effects on BAT metabolism.

### 3.5. PEMT Plays a Prominent Role in WAT during Aging

In contrast to BAT, PEMT deficiency had no strong impact on WAT metabolites (Figure 5A,C). In young mice, PEMT KO merely caused a decrease in acetate and glucose by less than 2-fold (Figure 5A, Appendix A). The separation of the two groups in the O-PLS-DA plot displayed a T[1] score of 18.7% (Figure 5B), and the significance of the clustering was addressed by a permutation test with an R^2^Y of 0.988 (*p* < 0.01) and a Q^2^ of 0.859 (*p* < 0.01) (Figure 5E).

As in plasma, most of the metabolic alterations were found in the WAT of old mice with an increase in threonine, serine, and arginine by less than twofold; a decrease in formate by more than threefold; and a decrease in glucose, glycerol, lactate, and acetate by more than twofold. The levels of 3-hydroxybutyrate and citrate were both decreased by less than twofold (Figure 5C, Appendix A). The O-PLS-DA resulted in a clustering of the two groups with a T[1] score of 38.2% (Figure 5D). The permutation test yielded an R^2^Y of 0.993 (*p* = 0.02) and a Q^2^ of 0.779 (*p* < 0.01) (Figure 5F).

The S-PLS-DA plot showed a distinct separation of all four groups, with the WT and KO groups aging differently (Figure 5G). This was further confirmed by the volcano plot of the WATs from the PEMT KO mice, which were dramatically different from those of the WT mice (Appendix A, [52]) This suggested an interplay between PEMT and aging in the regulation of WAT metabolism, with the role of PEMT gaining importance during aging. The ORA indicated that PEMT KO in the aged WAT affected the Warburg effect, glycine and serine metabolism, the transfer of acetyl groups into mitochondria, aspartate metabolism, gluconeogenesis, fatty acid biosynthesis, and galactose metabolism (Figure 5H).

### 3.6. Intestinal Segments Respond Differently to PEMT KO

The duodenum, the first part of the small intestine, showed an insensitive aging profile [52]; however, this changed with PEMT deficiency. In the tissues of young KO mice, we observed an increase in fumarate and a decrease in formate, choline, mannose, ethanolamine, glycine, lysine, lactate, and threonine (Figure 6A, Appendix A). Among these metabolites, only formate was decreased by more than twofold. The two groups were clustered on the O-PLS-DA plot with the score of the first predictive component being 31.2% (Figure 6B). In addition, the permutation yielded an R^2^Y at 0.895 (*p* = 0.02) and a Q^2^ at 0.681 (*p* = 0.01) (Figure 6E). The tissues of aged KO mice had a different set of altered metabolites, including increased dCTP, UDP-sugars, glucose, and guanosine and decreased uridine, formate, histidine, and ethanolamine (Figure 6C, Appendix A). Only dCTP increased more than twofold. The O-PLS-DA showed a T1 score of 15.6% (Figure 6D), and the permutation resulted in an R^2^Y of 0.997 (*p* < 0.01) and a Q^2^ of 0.892 (*p* < 0.01) (Figure 6F). On the S-PLS-DA plot, the two WT groups were clearly separated from the two KO groups, which was consistent with the volcano plot. Furthermore, the two KO groups were partly separated from each other with a greater difference between the young and aged PEMT KO mice compared to their WT counterparts (Figure 6G, Appendix A). Taken together, these results indicate that PEMT changes the metabolism of young and aged duodena in a distinct way, and its deficiency affects the metabolic stability of the duodenum. Enrichment analysis by the ORA revealed PEMT-associated pathways in the young duodenum such as the biosynthesis of various PLs and carnitine synthesis, (Figure 6H), whereas in the aged duodenum, the identified metabolites were related to lactose synthesis, methylhistidine metabolism, galactose metabolism, and sphingolipid metabolism (Figure 6I). The overall difference between the two sets of pathways clearly suggests a drastic change in the role of PEMT during duodenal aging.

Similar to the duodenum, the jejunum was shown in our previous studies as an “unaged” organ whose metabolic profile remained unchanged in old mice [52]. Here, we additionally demonstrated its insensitivity to PEMT deficiency (Figure 7A,C). The jejuna from young PEMT KO mice displayed solely an increase in fumarate (2.7-fold) and decreased levels of arginine and alanine by less than twofold (Figure 7A; Appendix A). The O-PLS-DA resulted in a clustering of the two groups with a T1 score of 26.4% (Figure 7B), and the permutation rendered an R^2^Y at 0.991 (*p* = 0.04) and a Q^2^ at 0.626 (*p* = 0.1), indicating an insignificant separation (Figure 7E). Uridine and inosine were increased by less than twofold in the jejuna of the aged KO mice (Figure 7C, Appendix A), and the O-PLS-DA of the two aged jejunum groups yielded a T[1] score of only 6.1%. The permutation test still indicated a significant separation of the two groups, with an R^2^Y at 0.992 (*p* = 0.01) and a Q^2^ at 0.707 (*p* = 0.01) (Figure 7F). The S-PLS-DA resulted in the overlap of the four clusters, (Figure 7G), which is consistent with a low level of metabolic changes in the aging and PEMT-deficient jejuna. In addition, we only observed increased uridine levels in the aging profile of the PEMT KO jejuna (Appendix A).

Surprisingly, PEMT deficiency strongly affected the metabolic profile of the ileum in the young mice. Therein, we found increased levels of glutamate, threonine, glycine, histidine, tryptophan, isoleucine, valine, phenylalanine, glycerol, tyrosine, hypoxanthine, asparagine, uracil, glutamine, leucine, methionine, and lysine by less than twofold, accompanied by decreased inosine (> fourfold) and fumarate (≈ twofold) (Figure 8A, Appendix A). The O-PLS-DA plot showed a separation of the two groups with a T[1] score of 38% (Figure 8B). The permutation test showed an R^2^Y at 0.976 (*p* < 0.01) and a Q2 at 0.923 (*p* < 0.01) (Figure 8E), further demonstrating a significant separation. The metabolism of aged KO tissues seems less affected, as the corresponding volcano plot showed only a slight increase in hypoxanthine and ethanolamine, and lower concentrations of allantoin, fumarate, dimethylamine, and uridine, with only allantoin showing a change greater than twofold (Figure 8C, Appendix A). The group clustering on the O-PLS-DA plot with a T[1] score of 14.5%, along with a permutation test resulting in an R^2^Y at 0.853 (*p* < 0.01) and a Q2 at 0.639 (*p* = 0.01), indicated a low predictive power despite a significant separation of the two groups (Figure 8D,F). The clusters of all four groups on the S-PLS-DA plot showed a consistent result with the marked metabolic alteration in the young tissues. In accordance, the enrichment analysis via an ORA revealed that the metabolic changes in the ilea of young PEMT KO mice compared to WT mice were associated with nine pathways (ammonia recycling, phenylalanine and tyrosine metabolism, purine metabolism, aspartate metabolism, urea cycle, glycine and serine metabolism, valine, leucine and isoleucine degradation, beta-alanine metabolism, and alanine metabolism), whereas no metabolic pathways were significantly associated with the metabolic change in the aged ilea of the PEMT KO mice in comparison to their WT counterparts. This indicates an essential role of PEMT in regulating metabolic homeostasis in the young ileum that decreases with aging. The aging of the KO ilea is associated with a significant decrease in the levels of multiple metabolites, which is in line with their increase in the young KO ilea (Appendix A).

To summarize all the differences in the metabolome between all the tissues studied, we created two heatmaps based on all the quantified metabolites (Figure 9, Appendix A). One displays the mean value of the metabolite abundance within each group (Figure 9), and the other displays all the relative abundances quantified from the NMR spectra (Appendix A). In line with previous observations [52], we found higher concentrations of amino acids in the metabolome of the three intestinal segments, and higher concentrations of tricarboxylic acid cycle (TCA cycle)- and lipid-associated metabolites in adipose tissues, as well as higher levels of amino acid derivatives in plasma and skeletal muscle. This further demonstrates the consistency of our data and the precision and robustness of the NMR-based metabolomic-profiling approach.

## 4. Discussion

In this study, the metabolomes of the liver, plasma, skeletal muscle, BAT, WAT, and intestine (duodenum, jejunum, and ileum) from young and aged WT and PEMT KO mice were investigated and revealed a plethora of metabolic changes due to the interference between aging and PEMT deficiency. The metabolomes of all PEMT KO tissues except the jejunum were strongly affected, demonstrating that global PEMT deficiency has profound systemic consequences. This finding was consistent with the importance of PC levels and PC/PE ratios in the individual tissues for maintaining the proper function of the liver, intestines, and skeletal muscle [19].

In recent decades, the consequences of PEMT deficiency have been extensively studied [39,40,43,60,61], but the underlying molecular mechanisms are still incompletely understood. Herein, we have provided the first systematic metabolomic characterization of both tissue-specific and age-dependent metabolic alterations in PEMT KO mice. Based on the outline of the metabolomic landscape uncovered in this study, we delineate (i) the metabolic changes due to the KO of PEMT, (ii) secondary effects to maintain tissue homeostasis and to cover the requirement of PC in response to the lack of functional PEMT, and (iii) indirect effects not directly linked to PL metabolism, but that are likely triggered by the availability of PCs and PEs. Most alterations in the tissue metabolomes of KO mice seemed to be indirect effects, for which an apparent link to PC biosynthesis could not be clearly established. The possible explanations for this involve the change in the PEMT-regulated protein expression level by alternative splicing [61] or through differed membrane fluidity due to PL compositions or acyl chain saturation levels [18,60,62].

Since PC is essential for the secretion of hepatic VLDL, the lack of choline leads to hepatic triacylglycerol (TG) accumulation [19,63,64]. Furthermore, a loss-of-function mutation of PEMT in humans may be associated with non-alcoholic fatty liver disease (NAFLD) as a possible consequence of altered PC synthesis and an increased TG accumulation [65]. This is consistent with previously identified hepatic steatosis in the PEMT KO mice [39]. Expectedly, we identified important changes in various metabolic pathways from the KO livers, which complements the previous mechanistic findings.

PEMT deficiency in the livers of young mice resulted in a drastic change of their metabolome, whereas the majority of the detected metabolites remained unchanged in the livers of the aged KO mice. Increased levels of AMP, histidine, inosine, acetic acid, propionate, and glutathione, as well as decreased phosphorylcholine and choline levels, were found to be comparable in both young and aged livers, suggesting that they are age-independent. This indicated that liver metabolism is largely stabilized during aging, possibly due to an increased supply of PC in aged mice.

Considering the changes in individual metabolites, the decrease in choline and phosphorylcholine are attributable to secondary effects of PEMT deficiency, likely due to a compensatory increase in PC synthesis through the CDP-choline pathway. Such a mechanism has already been identified in previous studies using both choline and phosphorylcholine as precursors of PCs [31,66]. The indirect consequences, such as increases in AMP in the livers of young and old KO mice, together with an increase in ADP in young KO mice, indicate a change in the ratio of ADP/ATP and AMP/ATP. This could result from an altered energy consumption. Interestingly, it has been previously found that PEMT deficiency results in an increased ATP concentration in hepatocytes [38], which contradicts our results. This discrepancy could be an outcome of changes in one or several factors of regulating ATP production, including Ca^2+^, ATP synthase, O_2_ flux, NADH/NAD^+^, uncoupling protein-2 (UCP2), or the citrate cycle [67,68,69], which are to be further explained to clarify the mechanistic impact of PEMT in liver energy production.

As for amino acids, histidine is expected to be catabolized into glutamate, with the latter being a precursor for glutathione biosynthesis [70,71]. Increased levels of both histidine and glutathione in young and aged mice could indicate alterations in this pathway. In addition, this might cause an increase in the glutamine concentration in young KO livers, which can be converted from glutamate [72]. However, further investigations are needed to clarify the exact changes that have taken place, and to determine whether an increased amino acid consumption is responsible for this and for the decrease in the glycine and phenylalanine levels.

The level of glycogen in KO livers has been found to decrease upon overnight fasting [38]. However, in the present study, we observed a reduced glucose level, but not glycogen, in the young KO livers. This indicates the absence or reduction in gluconeogenesis upon overnight fasting (see Methods) in the KO livers. Interestingly, the increase in acetate, which is likely a result of its reduced conversion to acetyl-CoA [73], could represent a compensatory mechanism to restore gluconeogenesis, as dietary acetate has been demonstrated to increase the glycogen concentration in rat livers [74]. We also found an increase in bile acids in young KO livers, whose main function is to facilitate digestion [75]. Overall, we observed the effects of PEMT deficiency on altering various biological functions in the liver, which could be a result of altered PL compositions in the different compartments of the liver such as cell membranes and VLDL.

One of the main roles of plasma is to transport nutrients and metabolites, whose metabolomes reflect various biological processes from different tissues [76,77]. The levels of plasma ethanolamine, choline, phosphorylcholine, and glycerophosphocholine were below the detection limits; therefore, the detected changes in the metabolome are mostly indirect consequences of PEMT deficiency. In young mice, the loss of PEMT solely led to an increase in plasma indoxyl sulfate, which is synthesized by cytochrome P450 and increases during aging or reduced renal clearance [52,78,79]. Interestingly, the plasma of aged KO mice showed opposite changes in the metabolites identified in aging mice [52], leading to the speculations that a lack of PEMT could delay or reverse the corresponding age-related alterations. Among the top five metabolic pathways associated with the altered plasma metabolome from the aged KO mice, four of them were related to aging biomarkers of WT plasma. These metabolic pathways are linked to myo-inositol, glucose, and glycerol. The decrease in myo-inositol in the plasma of the aged KO mice indicated an increased conversion to phosphatidylinositol-3-phosphate (PI3P) and increased autophagy [80,81]. Their elevated glucose levels are not in line with previous studies showing an improved insulin sensitivity and decreased hepatic gluconeogenesis in the KO mice [38,82]. This suggests a yet unknown mechanism promoting the production of glucose. The increase in free glycerol could be due to enhanced lipolysis during overnight fasting [83]. Citrate is a component of the TCA cycle and has been previously found to increase in aged WT plasma [52]. Its decrease in the aged plasma of KO mice suggests that age-associated lipid metabolism could be affected. Furthermore, circulating alanine levels increase in mice with hyperglycemia and may promote the elevation of plasma isoleucine concentrations [84]; we observed consistent results showing a concomitant increase in alanine, isoleucine, and glucose in old KO mice.

PEMT acts mainly in the liver and its expression in muscle tissues is extremely low [53,61]; however, a study demonstrated a phenotype of tissue-specific PEMT KO skeletal muscle with increased muscle oxygen consumption [60]. It is most likely that the observed metabolic changes in skeletal muscle are due to the indirect effects of PEMT deficiency. The previously described age-related metabolic phenotype of skeletal muscle from WT mice was consistent with decreased muscle function and activity [52]. Changes in the levels of metabolites including lactate, leucine, phenylalanine, and glucose during the aging of WT skeletal muscle [52] are not found in aged KO skeletal muscle (Appendix A). These metabolites are linked to the degree of energy consumption, whose absence among significantly altered metabolites may indicate a beneficial effect of PEMT deficiency on muscle aging via the functional recovery of muscles, as has been reported [60]. The resting metabolic rate declines with age [85]; an improved resting metabolic rate by PEMT suppression may reverse age-related metabolic alterations. Considering the conditions under which the metabolic rate should increase [60], a decrease in malonate might potentiate the consumption of succinate. Paradoxically, we observed elevated concentrations of succinate, suggesting an increased conversion of succinyl-CoA to succinate as a result of increased energy expenditure via the TCA cycle [86].

Taken together, PEMT deficiency increases the energy metabolism of skeletal muscle, which is known to decrease during aging [52]. However, the effects of aging and PEMT deficiency do not oppose and counteract each other, because PEMT also regulates metabolites whose concentrations do not change during aging. Therefore, skeletal muscle from aged PEMT KO mice is expected to have a higher functional activity than aged WT mice. Together with the observations in the metabolome of aged plasma, more studies are required to examine whether a phenotype showing an age-delayed effect in KO mice is in place.

Brown adipocytes are rich in lipid droplets (LDs) and mitochondria, and contact between LDs and mitochondria in these adipocytes could take place to facilitate fatty acid oxidation [87,88,89]. In a previous study, it has been demonstrated that PEMT is required for the formation of stable LDs, whose absence causes basal lipolysis in mouse WAT and 3T3-L1 adipocytes [21]. The lack of PEMT was shown not to impair the thermogenic capacity of BAT [42]; however, both our results and a previous report [61] have suggested dramatic metabolic alterations of BAT upon PEMT deficiency. Of note, despite the fact that the expression of PEMT in BAT has been confirmed by mRNA quantification and immunoblotting [53,61], its function has been considered liver-specific and most functions of BAT are not directly regulated by locally expressed PEMT [31,61]. If the PEMT expressed in BAT directly converts local PE to PC, tissue homeostasis responding to PEMT deficiency might increase PC synthesis via the CDP-choline pathway or decrease PC metabolism, as well as increase PE metabolism via phospholipase activities [31,90,91], causing a decrease in choline and an increase in ethanolamine levels. However, we only observed a decrease in ethanolamine in the KO BAT, whereas the choline levels remained unchanged. However, in both young and aged KO BAT we observed a decrease in glycerophosphocholine, which is a potential source of choline in yeast [92] and the product of PC catabolism in various cells [93,94]. This could indicate that a homeostatic process increasing PC production has taken place, but through the consumption of glycerophosphocholine instead of choline.

As for the indirect effect of PEMT deficiency in BAT, the decrease in the levels of metabolites from KO BAT manifest in an age-independent manner (Appendix A), and most aging biomarkers whose levels are increased during aging are found in both WT and KO BAT. This might result from altered lipolysis due to destabilized LDs, since both BAT and WAT contain LDs and can be regulated by PEMT in a comparable manner [21,89]. However, most changes in the metabolome of BAT caused by PEMT deficiency are independent of age and vice versa. Among the decreased metabolites in the KO BAT, BCAAs are known to supply BAT thermogenesis, and glutamate is a byproduct of BCAA catabolism [95,96]. Taken together with the decrease in acetate, which is an inhibitor of thermogenesis in BAT [97], our results indicate a slight increase in energy consumption in the KO BAT. Together with the increased citrate concentration in the BAT of young mice, this is in line with the altered mitochondrial function of KO mice as previously identified [38]. More experiments quantifying the energy consumption and LD formation of BAT would be necessary to validate the significance of these observations.

WAT is responsible for lipid storage and endocrine secretion; it may acquire a thermogenic function upon browning, and our previous studies have shown that its metabolism changes little during aging [52,98,99]. PEMT deficiency has been found to result in decreased lipogenesis, which may explain the resistance to high fat diet-induced obesity [62]. We were only able to detect metabolic changes in aged PEMT KO mice, indicating that only combined aging and PEMT deficiency can significantly alter the abundance of several metabolites. Upon the decreased lipogenesis that occurs in WAT, the decrease in glucose in both young and aged KO WAT may be a result of reduced gluconeogenesis or an increased glucose utilization [38]. In agreement, the increase in serine in the WAT from aged KO mice could indicate an increased conversion from glucose [100]. In the metabolome of WAT from aged PEMT KO mice, the decreased levels of glycerol could be one of the reasons for the decreased degree of lipogenesis [62]. Increased amounts of glycerol in plasma can be released from WAT as a product of the elevated basal lipolysis of triglycerides [21,83]. Citrate is a component of the TCA cycle [73] whose decrease may be due to an increased flux through the TCA cycle [38], in combination with a higher conversion of acetyl-CoA. This is consistent with the decrease in glucose and acetate, which are the possible precursors of acetyl-CoA [101]. Interestingly, the concentration of citrate is increased in the plasma of aged KO mice, as well as in young KO BAT, but remains unchanged in aged KO BAT. This could result from the export of citrate in the plasma from both BAT and WAT in aged KO mice. Overall, the observations in WAT are consistent with decreased levels of lipogenesis and fat accumulation, which is in line with previous studies [62].

Biliary PC solubilizes dietary lipids in the small intestine and facilitates their absorption [19,102]. The secretory rate of biliary PC does not differ between WT and PEMT KO mice [103]. However, PEMT KO affects Na^+^ absorption and subsequent bile acid re-absorption in the small intestine [104] by an as yet unknown mechanism, and biliary PC may be important for choline homeostasis [105]. Despite its involvement in nutrient absorption, our previous study identified the duodenum as a tissue without significant metabolic alterations during aging [52]. Here, we observed only age-dependent metabolic changes in the duodenum of the PEMT KO mice. It has been found that the small intestine acquires PC from the diet based on a rat model [106]. Thus, our results may indicate the usage of non-negligeable amounts of PC produced by the liver in the duodenum, if not the entire small intestine. PEMT deficiency, therefore, may change the PC supply with respect to both the quantities and types of fatty acids during aging [35]. In comparison to the duodena from young WT mice, young KO animals exhibit decreased duodenal levels of choline and ethanolamine. We have identified that in both young and old mice, the loss of PEMT caused a decrease in the levels of formate and ethanolamine, which are both related to the gut microbiome [107,108,109]. The highest difference in the abundance of metabolites between young and old PEMT KO mice was seen for glucose, UDP-sugars, and mannose, indicating the indirect, age-related effects of PEMT deficiency on carbohydrate metabolism.

We observed only modest alterations in the jejunal metabolome caused by the loss of PEMT, showing its stable metabolism. Correspondingly, we have previously reported the jejunum as a tissue that maintains stable tissue homeostasis throughout a lifetime in mice [52]. The levels of multiple components, including fumarate, uridine, ethanolamine, glycine, lysine, threonine, histidine, and inosine, were reversely altered in the different segments of the small intestine. This could originate from unknown changes in the cellular metabolism, nutritional sources, or the metabolic activity of the microbiome.

Surprisingly, the ileum was one of the tissues profoundly affected by PEMT deficiency, predominantly in young mice. We observed an increase in the levels of almost all amino acids, indicating increased nutrient absorption due to changes in cellular metabolism. In both young and aged KO tissues, we identified increased concentrations of hypoxanthine, a metabolic product of inosine [110], which was concomitantly reduced in the ilea of young PEMT KO mice. As the metabolome of the WT ileum also displays only few changes during aging [52], a differentiated liver-originated PC supply throughout the aging process may affect all three segments of the small intestine, causing remarkable changes in the metabolome of duodenum and ileum, and fewer changes in the jejunum. Fumarate, an intermediate of the TCA cycle [73], was decreased in the PEMT KO ileum, which might have been a response to altered nutrient absorption and may contribute to the recovery of normal metabolism. Active PEMT has been detected in the small intestine of rats [111], but studies on the phenotype of the KO small intestine are scarce. Therefore, further research is needed to validate the physiological significance of our findings.

Overall, our studies revealed a regulatory role of PEMT in the metabolism of various tissues, mainly through the TCA cycle and amino acid catabolism. Consequently, it manifests partly in the reversal of the aging-associated metabolic alterations in plasma and skeletal muscle, and as age-dependent homeostatic regulation in adipose tissues and intestines. These key findings have been summarized in Figure 10. Future investigations should focus on a specific tissue to demonstrate PEMT-associated pathways to reveal the change in the PL composition and its biological consequences.

## 5. Conclusions

During this study, we performed NMR-based analyses to investigate the impact of PEMT deficiency on the metabolomes of the livers, plasma, skeletal muscle, adipose tissue, and small intestines from young and old mice. We identified diverse consequences in different tissues due to either the alterations in lipid uptake and transport, the compensation for the altered PC production, or various functional changes due to altered PL compositions. These responses were either age-dependent or age-independent and underline the complex metabolic role of PEMT.

## Figures and Tables

**Figure 1 biomolecules-12-01270-f001:**
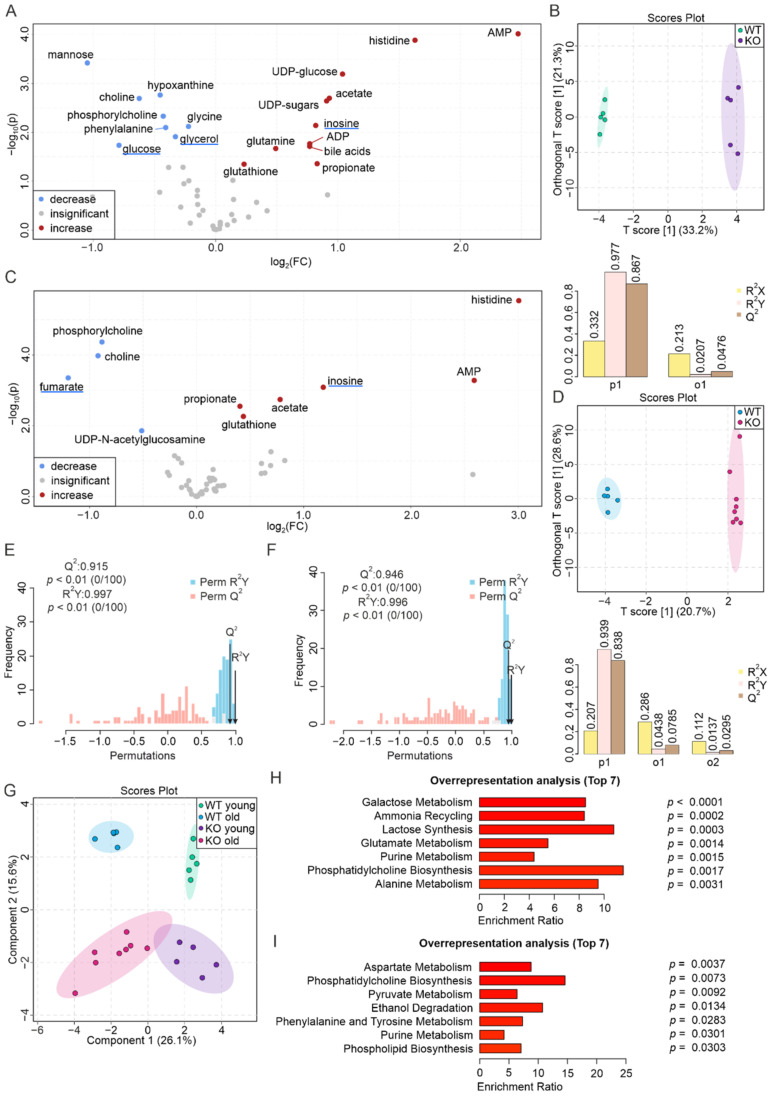
Drastic alteration of liver metabolome as a consequence of PEMT deficiency. (**A**) Volcano plot showing the differences in metabolite levels between WT and PEMT KO livers from young mice (normalization was performed for each pairwise comparison). Red and blue dots represent significantly increased and decreased metabolites (*p* < 0.05), respectively, whereas grey dots represent metabolites with insignificant changes (*p* > 0.05). Blue and red underlines indicate aging biomarker(s) whose concentration(s) decrease(s) and increase(s) in the WT aged liver, respectively, as well as in (C). (**B**) O-PLS-DA plot of WT (*n* = 5, green) and KO (*n* = 5, purple) liver samples from young mice (upper panel) and cross-validation coefficients R2X, R2Y, and Q2 of the first predictive component (p1) and the first orthogonal component (o1) (lower panel). (**C**) Volcano plot showing differences in the abundance of metabolites between WT and PEMT KO livers from aged mice. (**D**) O-PLS-DA plot of WT (*n* = 5, blue) and KO (*n* = 8, pink) liver samples from aged mice (upper panel) and model quality assessment for p1 and o1 with cross-validation coefficients R^2^X, R^2^Y, and Q^2^ (lower panel). (**E**) Histogram for the permutation test of the samples in (**B**) and (**F**) of the samples in (D) with permutation number *n* = 100, respectively. (**G**) S-PLS-DA plot showing the clustering of samples from the four liver groups, with green, blue, purple, and pink corresponding to WT young, WT old, KO young, and KO old, respectively. (**H**) ORA of top 7 pathways associated with significantly altered metabolites in livers of young mice (**A**), and (**I**) ORA of top 7 pathways associated with significantly altered metabolites in aged mice (**C**) with all pathways yielding *p* < 0.05.

**Figure 2 biomolecules-12-01270-f002:**
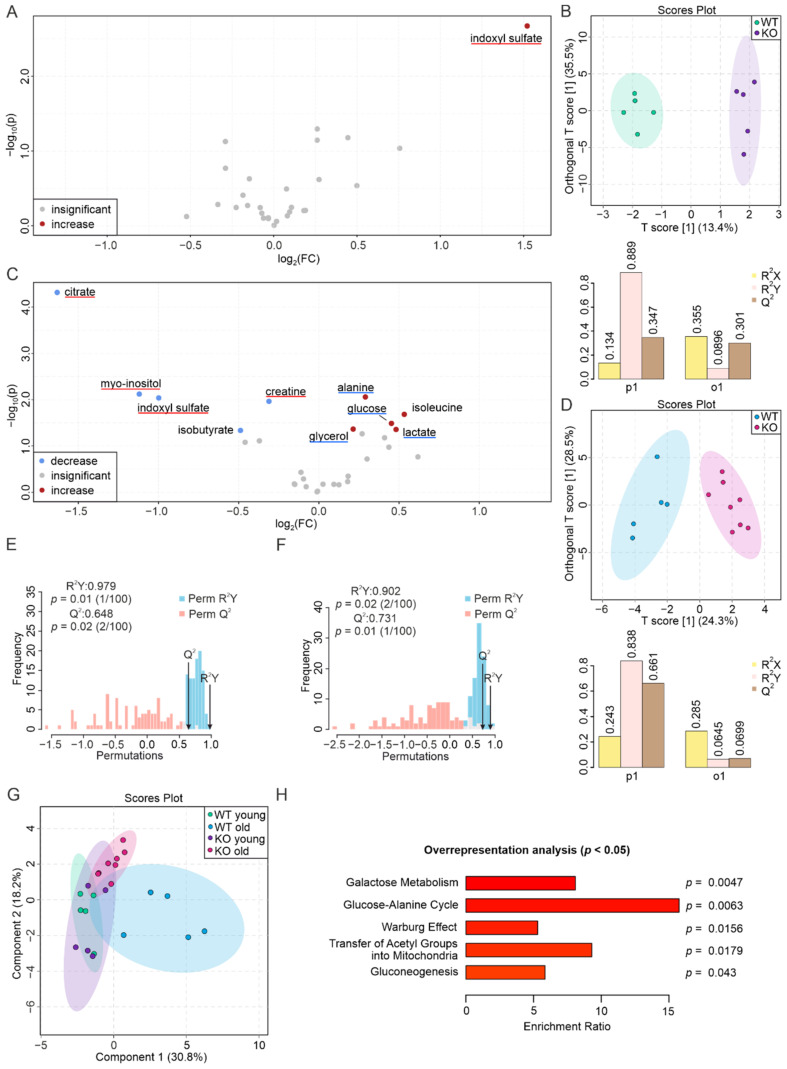
PEMT deficiency affects plasma metabolites upon aging. (**A**) Volcano plot showing differences in the abundance of metabolites between WT and PEMT KO plasma from young mice. Red dot represents a significantly increased metabolite (*p* < 0.05). Grey dots denote metabolites with insignificant changes (*p* > 0.05). Blue and red underlines indicate aging biomarker(s) whose concentration(s) decrease(s) and increase(s) in the WT aged plasma, respectively, as well as in (**C**). (**B**) O-PLS-DA plot of WT (*n* = 5, green) and KO (*n* = 5, purple) plasma samples from young mice (upper panel), and model quality assessment for p1 and o1 with cross-validation coefficients R^2^X, R^2^Y, and Q^2^ (lower panel). (**C**) Volcano plot showing differences in the abundance of metabolites between WT and PEMT KO plasma from aged mice. Red and blue dots correspond to significantly increased and decreased metabolites, respectively. Grey dots denote metabolites with insignificant changes. (**D**) O-PLS-DA plot of WT (*n* = 5, blue) and KO (*n* = 8, pink) plasma samples from aged mice (upper panel), and model quality assessment for p1 and o1 with cross-validation coefficients R^2^X, R^2^Y, and Q^2^ (lower panel). (**E**) Histogram showing the permutation test of samples in (**B**) and (**F**) samples in (**D**) with permutation number *n* = 100, respectively. (**G**) S-PLS-DA plot showing the clustering of samples from the 4 plasma groups, with green, blue, purple, and pink corresponding to WT young, WT old, KO young, and KO old, respectively. (**H**) ORA of pathways associated with significantly altered metabolites in the plasma of aged mice (**C**), with all pathways yielding *p* < 0.05.

**Figure 3 biomolecules-12-01270-f003:**
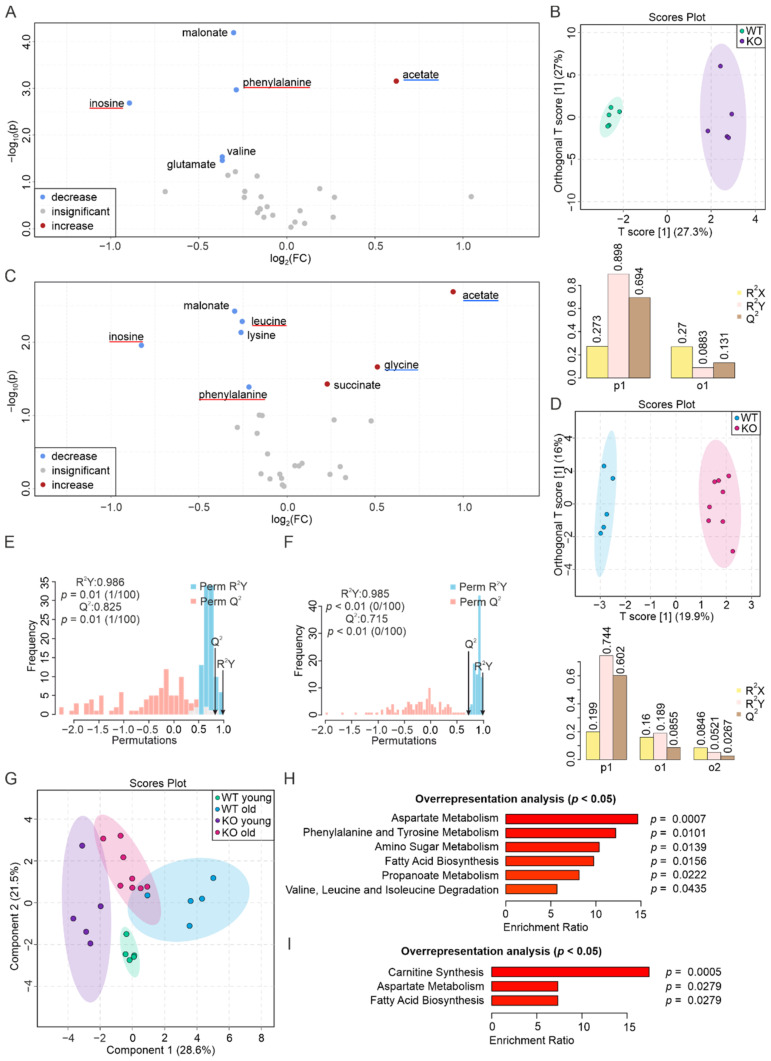
Contrasting metabolic changes in skeletal muscle upon PEMT deficiency and aging. (**A**) Volcano plot indicating increased (red), insignificantly changed (grey) and decreased (blue) metabolites in the PEMT KO skeletal muscle of young mice, in comparison to their WT counterparts. Underlines indicate aging biomarker(s) whose concentration(s) decrease(s) (blue) and increase(s) (red) in the WT aged skeletal muscle, as well as in (**C**). (**B**) O-PLS-DA plot clustering WT (*n* = 5, green) and KO (*n* = 5, purple) skeletal muscle samples from young mice (upper panel), along with the model quality assessment for p1 and o1 with cross-validation coefficients R^2^X, R^2^Y, and Q^2^ (lower panel). (**C**) Volcano plot indicating increased (red), insignificantly changed (grey), and decreased (blue) metabolites in the PEMT KO skeletal muscle of aged mice, in comparison to their WT counterpart. (**D**) O-PLS-DA plot clustering WT (*n* = 5, blue) and KO (*n* = 8, pink) skeletal muscle samples from aged mice (upper panel), and model quality assessment for p1, o1, and o2 (second predictive component) with cross-validation coefficients R^2^X, R^2^Y, and Q^2^ (lower panel). (**E**) Histogram showing the permutation test of samples in (**B**) and (**F**) samples in (**D**) with permutation number *n* = 100, respectively. (**G**) S-PLS-DA plot of 4 skeletal muscle groups, with green, purple, blue, and pink corresponding to WT young, KO young, WT old, and KO old, respectively. (**H**) ORA of pathways associated with significantly changed metabolites in the skeletal muscle of young mice (**A**), and (**I**) of aged mice (**C**), with all pathways yielding *p* < 0.05.

**Figure 4 biomolecules-12-01270-f004:**
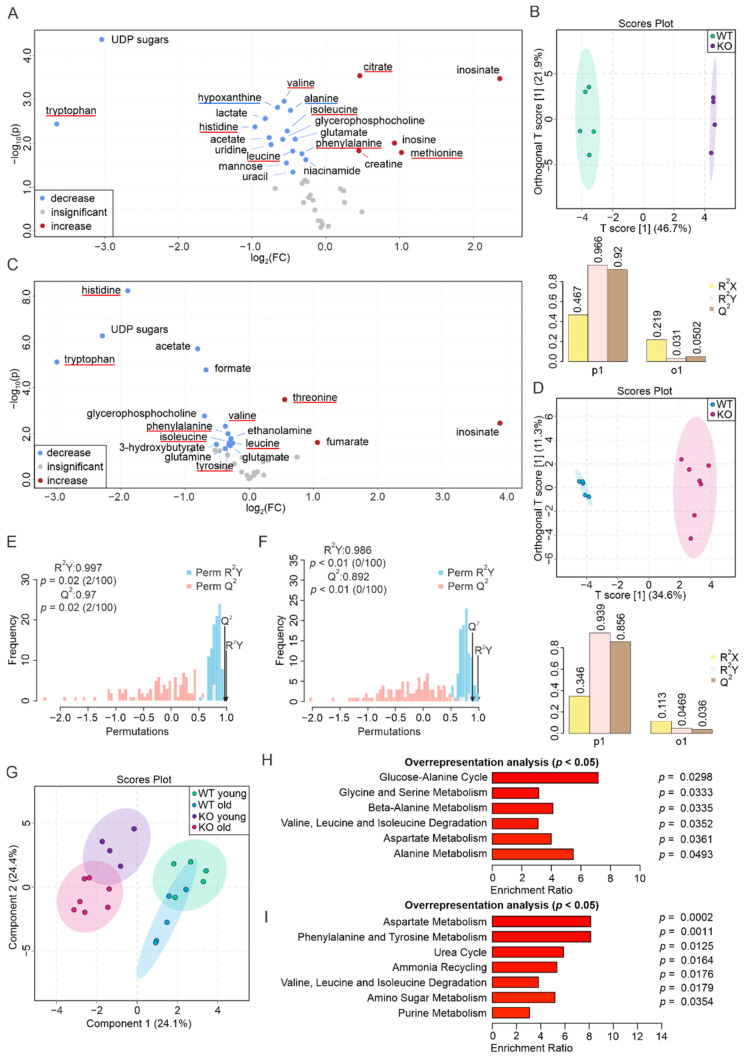
Metabolic alterations in the PEMT KO BAT from young and aged mice. (**A**) Volcano plot showing differences in the metabolite abundance between WT and PEMT KO BAT from young mice. Red and blue dots correspond to increased and decreased metabolites, respectively. Grey dots denote metabolites with insignificant changes. Blue and red underlines indicate aging biomarker(s) whose concentration(s) decrease(s) and increase(s) in the WT aged BAT, respectively, as well as in (**C**). (**B**) O-PLS-DA plot of WT (*n* = 5, green) and KO (*n* = 4, purple) BAT samples from young mice (upper panel), and cross-validation coefficients R^2^X, R^2^Y, and Q^2^ of p1 and o1 (lower panel). (**C**) Volcano plot showing differences in the abundance of metabolites between WT and PEMT KO BAT from aged mice. (**D**) O-PLS-DA plot (upper panel) and cross-validation coefficients R^2^X, R^2^Y, and Q^2^ of p1 and o1 (lower panel) of WT (*n* = 5, blue) and KO (*n* = 7, pink) BAT samples from aged mice. (**E**) Histogram of permutation test for samples in (**B**) and (**F**) samples in (**D**) with permutation number *n* = 100, respectively. (**G**) S-PLS-DA plot of the 4 BAT groups, with green, purple, blue, and pink corresponding to WT young, KO young, WT old, and KO old, respectively. (**H**) ORA of pathways associated with significantly changed metabolites in the BAT of young mice (**A**) and (**I**) of aged mice (**C**), with all pathways yielding *p* < 0.05.

**Figure 5 biomolecules-12-01270-f005:**
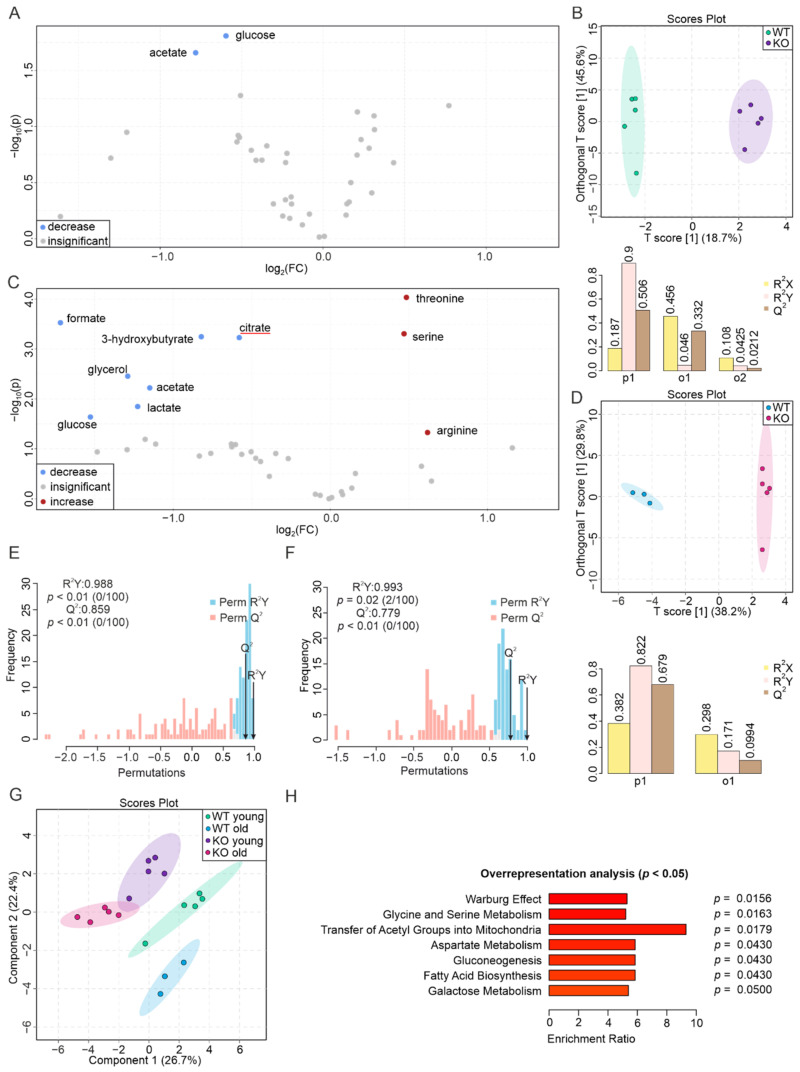
Metabolic alterations in WAT of young and aged PEMT KO mice. (**A**) Volcano plot showing differences in the metabolic profiles of WT and PEMT KO WAT from young mice. Red, blue, and grey dots indicate significantly increased, decreased, and insignificantly changed metabolites, respectively. (**B**) O-PLS-DA plot of WT (*n* = 5, green) and KO (*n* = 5, purple) WAT samples from young mice (upper panel), and model quality assessment for p1, o1, and o2 with cross-validation coefficients R^2^X, R^2^Y, and Q^2^ (lower panel). (**C**) Volcano plot showing differences in the metabolic profiles of WT and PEMT KO WAT from aged mice. Red underline indicates an aging biomarker whose concentration increases in the aged WT WAT. (**D**) O-PLS-DA plot of WT (*n* = 3, blue) and KO (*n* = 5, pink) WAT samples from aged mice (upper panel), and model quality assessment for p1 and o1 with cross-validation coefficients R^2^X, R^2^Y, and Q^2^ (lower panel). (**E**) Permutation test of samples in (**B**) and (**F**) samples in (**D**) with permutation number *n* = 100, respectively. (**G**) S-PLS-DA plot of the 4 WAT groups, with green, purple, blue, and pink corresponding to WT young, KO young, WT old, and KO old, respectively. (**H**) ORA of pathways associated with significantly changed metabolites in the WAT of aged mice (**C**), with all pathways yielding *p* < 0.05.

**Figure 6 biomolecules-12-01270-f006:**
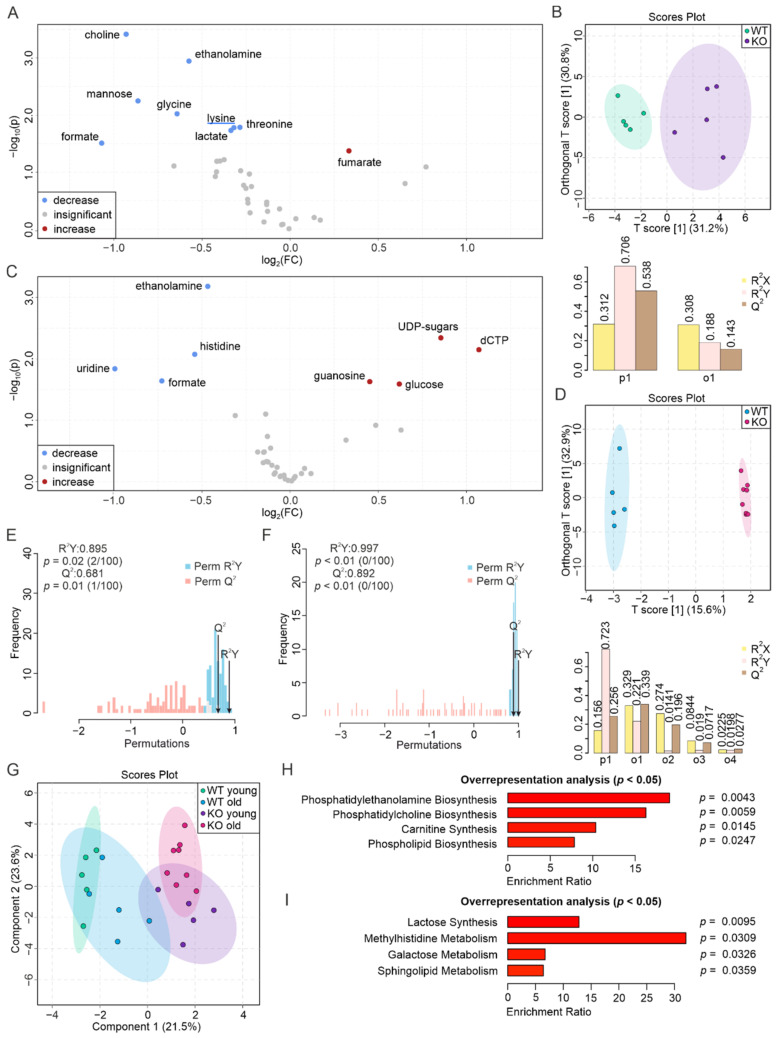
PEMT deficiency affects metabolic stability of young and old duodena. (**A**) Volcano plot showing differences in the metabolite concentrations between WT and PEMT KO duodenum from young mice. Red and blue corresponds to significantly increased and decreased metabolites, respectively. Grey dots denote metabolites with insignificant changes. Blue underline indicates an aging biomarker whose concentration decreases in the aged WT duodenum. (**B**) O-PLS-DA plot of WT (*n* = 5, green) and KO (*n* = 5, purple) duodenum samples from young mice (upper panel), and model quality assessment for p1 and o1 with cross-validation coefficients R^2^X, R^2^Y, and Q^2^ (lower panel). (**C**) Volcano plot showing differences in the metabolite concentrations between WT and PEMT KO duodena from aged mice. (**D**) O-PLS-DA plot of WT (*n* = 5, blue) and KO (*n* = 8, pink) duodenum samples from aged mice (upper panel), and model quality assessment for p1 and four orthogonal components (o1, o2, o3, and o4) with cross-validation coefficients R^2^X, R^2^Y, and Q^2^ (lower panel). (**E**) Histogram of permutation test for samples in (**B**) and (**F**) for samples in (**D**) with permutation number *n* = 100, respectively. (**G**) S-PLS-DA plot of the 4 duodenum groups, with green, purple, blue, and pink corresponding to WT young, KO young, WT old, and KO old, respectively. (**H**) ORA of pathways associated with significantly changed metabolites in the duodenum of (**A**) young mice and (**I**) of aged mice (**C**), with all pathways yielding *p* < 0.05.

**Figure 7 biomolecules-12-01270-f007:**
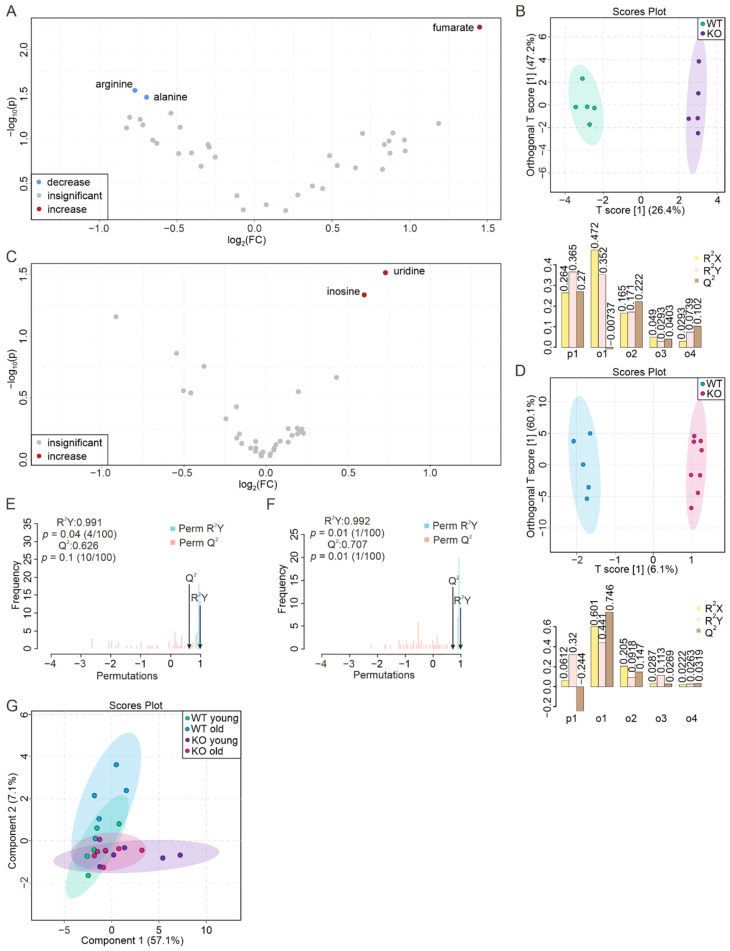
Jejunal metabolites are hardly affected by ageing or PEMT deficiency. (**A**) Volcano plot showing differences in the metabolic profiles of WT and PEMT KO jejuna from young mice. Red and blue corresponds to significantly increased and decreased metabolites, respectively. Grey dots correspond to metabolites with insignificant changes. (**B**) O-PLS-DA plot of WT (*n* = 5, green) and KO (*n* = 5, purple) jejunum samples from young mice (upper panel), and model quality assessment for p1 and four orthogonal components (o1, o2, o3, and o4) with cross-validation coefficients R^2^X, R^2^Y, and Q^2^ (lower panel). (**C**) Volcano plot showing differences in the metabolic profiles of WT and PEMT KO jejuna from aged mice. Red dots represent significantly increased metabolites. Grey dots denote metabolites with insignificant changes. (**D**) O-PLS-DA plot of WT (*n* = 5, blue) and KO (*n* = 8, pink) jejunum samples from aged mice (upper panel), and model quality assessment for p1 and four orthogonal components (o1, o2, o3, and o4) with cross-validation coefficients R^2^X, R^2^Y, and Q^2^ (lower panel). (**E**) Histogram showing permutation test of samples in (**B**) and (**F**) of samples in (**D**) with permutation number *n* = 100, respectively. (**G**) S-PLS-DA plot of the 4 jejunum groups, with green, purple, blue, and pink corresponding to WT young, KO young, WT old, and KO old, respectively.

**Figure 8 biomolecules-12-01270-f008:**
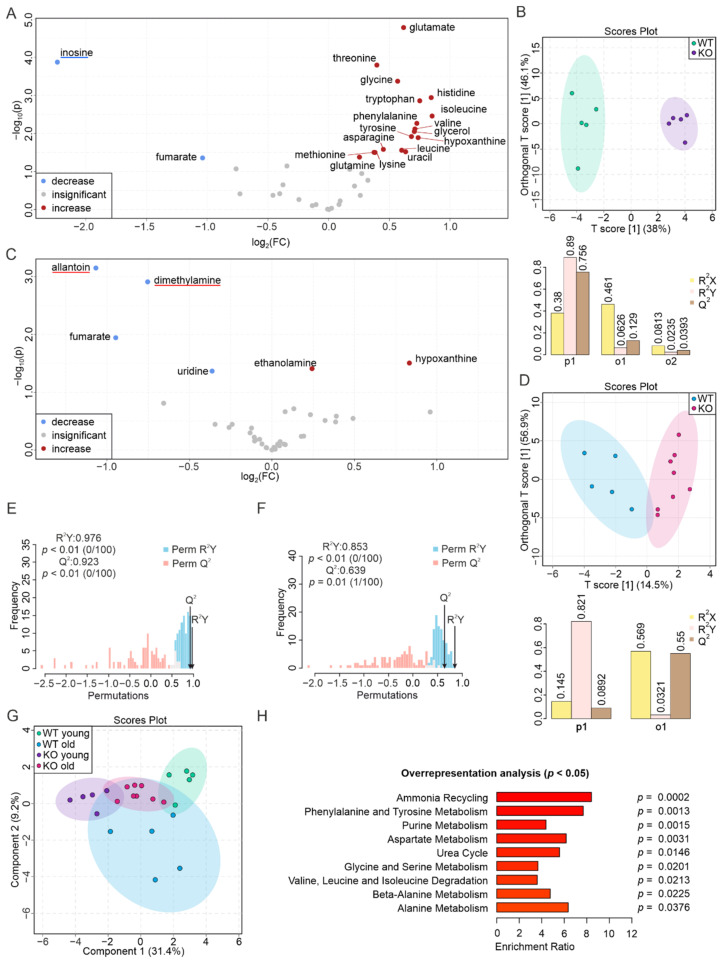
PEMT deficiency severely affects the metabolome of young ilea. (**A**) Volcano plot showing differences in the metabolite concentrations between WT and PEMT KO ilea from young mice. Red, blue, and grey dots correspond to significantly increased, decreased, and insignificantly changed metabolites, respectively. Blue underline indicates an aging biomarker whose concentration decreases upon aging in the WT ileum. (**B**) O-PLS-DA plot of WT (*n* = 5, green) and KO (*n* = 5, purple) ileum samples from young mice (upper panel), and cross-validation coefficients R^2^X, R^2^Y, and Q^2^ of p1, o1, and o2 (lower panel). (**C**) Volcano plot showing differences in metabolite concentrations between WT and PEMT KO ilea from aged mice. Red underlines highlight aging biomarkers whose concentrations increase upon aging in WT ilea. (**D**) O-PLS-DA plot of WT (*n* = 5, blue) and KO (*n* = 8, pink) ileum samples from aged mice (upper panel), and cross-validation coefficients R^2^X, R^2^Y, and Q^2^ of p1 and o1 (lower panel). (**E**) Histogram showing permutation test of samples in (**B**) and (**F**) of samples in (**D**) with permutation number *n* = 100, respectively. (**G**) S-PLS-DA plot of the 4 ileum groups, with green, purple, blue, and pink corresponding to WT young, KO young, WT old, and KO old, respectively. (**H**) ORA of pathways associated with significantly changed metabolites in the ileum of young mice (**A**), with all pathways yielding *p* < 0.05.

**Figure 9 biomolecules-12-01270-f009:**
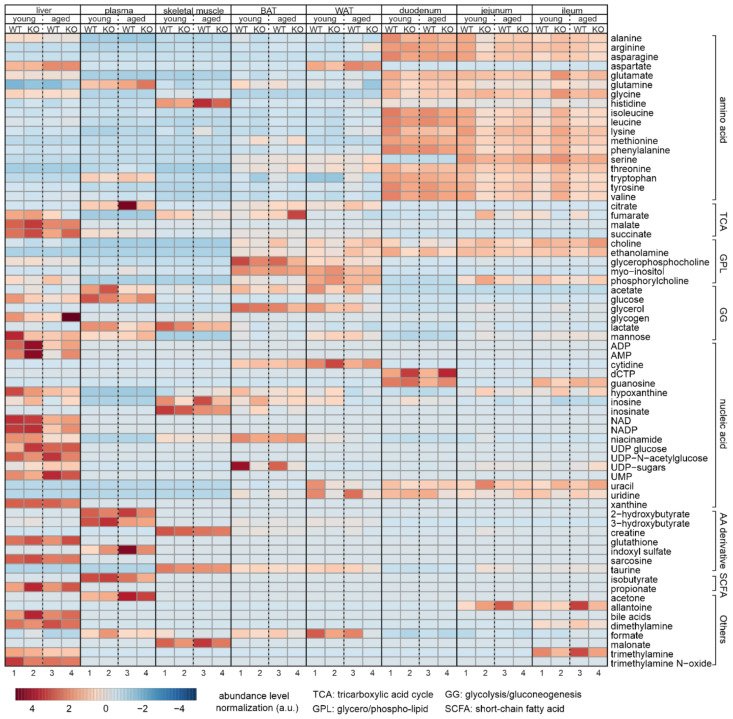
Heatmap showing the average of quantified metabolites within each group of tissues. Type of tissue, status (young or aged), and genotypes are indicated on the top, and numbers of group are shown below. Each line corresponds to a specific metabolite labeled on the right. Each single cell represents the average value calculated from all the relative abundances of the corresponding metabolites within the group. Each of the 4 groups from the same tissue are normalized to enable a comparison. Metabolites at higher concentrations and lower concentrations are in red and blue, respectively.

**Figure 10 biomolecules-12-01270-f010:**
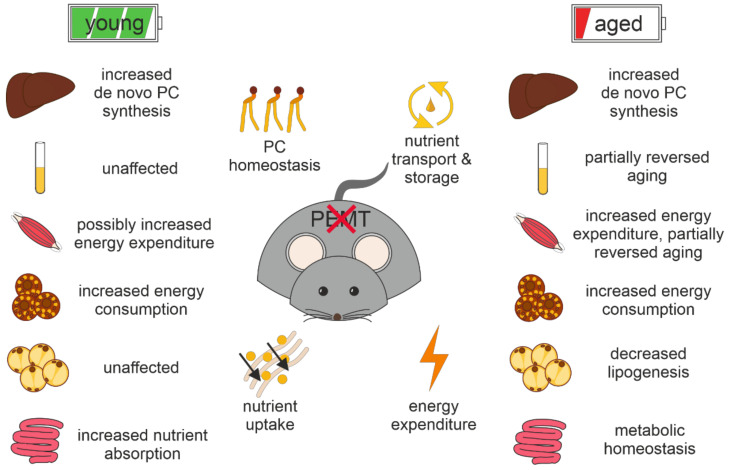
Summary of the key findings from untargeted metabolomic investigation of PEMT KO mouse tissues.

## Data Availability

The data presented in this study are available on request from the corresponding author.

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
