# Peer review of "Phosphatidylethanolamine *N*-Methyltransferase Knockout Modulates Metabolic Changes in Aging Mice"

_biomolecules, 2022, doi:10.3390/biom12091270_

Round 1
Reviewer 1 Report
The authors performed NMR-based metabolomics analysis in several tissues in WT and PEMT KO mice at different age. Their analysis found that PEMT deficiency affect the metabolism of glucose, amino acid, and fatty acid. Especially, the alterations of liver and BAT were dramatic. The obtained data could contribute to understand the importance of phospholipid in aging-dependent metabolomics changes. They stated some important possibilities in discussion section, however the reviewer think additional experiment is necessary to indicate above possibilities.
1. The authors indicate some possibilities based on hepatic metabolome data, including glutathione, and UCP2. And, from plasma and BAT data, they suggest PEMT could delay the age-related alteration, and its KO increase energy consumption, respectively. However, there is no phenotype or its-related data to support their possibilities. I requested some additional data.
2. In many tissues, the metabolites of TCA cycle, amino acids, glycolysis were altered by PEMT KO. As the authors described that PEMT is abundant in hepatic tissue, these global alteration is interesting. How these changes were induced by PEMT KO? In second paragraph of discussion section, they descried some statement regarding this point. But, please state more detail.
3. Do you have some basic information such as body weight, tissue weight, and food intake?
4. In this study, the authors used female mice. Why chose female mice??
5. In metabolome analysis, they used aging biomarkers. How these markers are defined??
And how many metabolites are detected authors’ metabolome analysis??
Author Response
Question 1: The authors performed NMR-based metabolomics analysis in several tissues in WT and PEMT KO mice at different age. Their analysis found that PEMT deficiency affect the metabolism of glucose, amino acid, and fatty acid. Especially, the alterations of liver and BAT were dramatic. The obtained data could contribute to understand the importance of phospholipid in aging-dependent metabolomics changes. They stated some important possibilities in discussion section, however the reviewer think additional experiment is necessary to indicate above possibilities.
Answer 1: The authors carefully rechecked their interpretations in order to provide more coherent statements in the discussion part and removed claims that might create ambiguity or over interpretations.
Question 2: The authors indicate some possibilities based on hepatic metabolome data, including glutathione, and UCP2. And, from plasma and BAT data, they suggest PEMT could delay the age-related alteration, and its KO increase energy consumption, respectively. However, there is no phenotype or its-related data to support their possibilities. I requested some additional data.
Answer 2: The authors replaced the UCP2 statement by invoking further studies on the mechanisms causing altered ATP production in KO liver. The same simplification has been applied to glutathione.
Regarding the phenotypes of tissues from PEMT KO mice, previous studies revealed phenotypes of liver, skeletal muscle, BAT and WAT with PEMT deficiency. These phenotypes provided important information to understand our data. In addition to referring and citing these studies, we have now included additional statements. If information regarding the phenotypes was scarce, we have also added a statement that additional investigations are necessary to validate the findings.
Question 3: In many tissues, the metabolites of TCA cycle, amino acids, glycolysis were altered by PEMT KO. As the authors described that PEMT is abundant in hepatic tissue, these global alteration is interesting. How these changes were induced by PEMT KO? In second paragraph of discussion section, they descried some statement regarding this point. But, please state more detail.
Answer 3: Based on current knowledge, a relationship between PEMT deficiency and altered energy expenditure could be established. However, the molecular mechanisms are still unclear. Possible explanations could be regulation of gene expression and membrane fluidity, which the authors added at the end of the second paragraph in the discussion.
Question 4: Do you have some basic information such as body weight, tissue weight, and food intake?
Answer 4: The mice in this experiment have been fed with chow diet ad libitum. We did not measure body weight and tissue weights of the mice as the original publication (https://doi.org/10.1073/pnas.94.24.12880) demonstrated comparable body weight of Pemt-ko and wild-type mice.
Question 5: In this study, the authors used female mice. Why chose female mice??
Answer 5: We chose female mice since females show a more striking decrease in the amount of plasma phosphatidylcholine and cholesterol in high density lipoproteins than male Pemt-ko mice (https://doi.org/10.1074/jbc.M301982200).
Question 6: In metabolome analysis, they used aging biomarkers. How these markers are defined??
Answer 6: Aging biomarkers are defined as metabolites whose concentrations are significantly increased or decreased (p < 0.05) in the aged mice tissues, with p values determined by unpaired two-tailed Student’s t-test based on the quantified individual metabolite concentrations from young and aged tissues, as described in the Materials and Methods.
Question 7: And how many metabolites are detected authors’ metabolome analysis??
Answer 7: The number of detected metabolites is tissue dependent. For example, in liver, 48 metabolites have been identified, which is the most among all tissues. 29 metabolites were detected in plasma, 27 in skeletal muscle, 40 in BAT, 38 in WAT, 36 in duodenum, 36 in jejunum and 38 in ileum, respectively.
Reviewer 2 Report
In this manuscript, the authors evaluated the consequences of PEMPT knockout in different tissues by metabolomic approach. The manuscript is well written and results may be of interest for researchers working on metabolism. A limitation of the study is represented by the descriptive nature of the work, which is, however, well counterbalanced by an extensive integration with the state of the art. I suggest to provide tables/working models to summarize the main findings.
Author Response
In this manuscript, the authors evaluated the consequences of PEMPT knockout in different tissues by metabolomic approach. The manuscript is well written and results may be of interest for researchers working on metabolism. A limitation of the study is represented by the descriptive nature of the work, which is, however, well counterbalanced by an extensive integration with the state of the art. I suggest to provide tables/working models to summarize the main findings.
The authors thank the reviewer for her/his positive evaluation of our manuscript. To address the suggestion, the authors added a graphic summary in Figure 10.
Reviewer 3 Report
In this manuscript, the effects of phosphatidylethanolamine N-methytransferase knockout in modulating metabolic changes with age using mouse models. It is shown that the effect is both age and tissue specific - but affecting all tissues. It is also shown that PEMT KO triggers complex changes as shown by increase in skeletal muscle energy metabolism (even in old age) while not affecting thermogenic capacity of brown adipose tissue.
Overall, the analysis is carefully carried out and has produced results that are very interesting.
Author Response
In this manuscript, the effects of phosphatidylethanolamine N-methytransferase knockout in modulating metabolic changes with age using mouse models. It is shown that the effect is both age and tissue specific - but affecting all tissues. It is also shown that PEMT KO triggers complex changes as shown by increase in skeletal muscle energy metabolism (even in old age) while not affecting thermogenic capacity of brown adipose tissue.
Overall, the analysis is carefully carried out and has produced results that are very interesting.
The authors thank the reviewer for her/his positive evaluation of our manuscript.
Round 2
Reviewer 1 Report
The authors modified the discussion section adequately, inserted the reference, and include their previous works. The further work is required in this animal model, however, the reviewer consider the revised manuscript could be acceptable for the publication.